# Oxidative Stress, Environmental Pollution, and Lifestyle as Determinants of Asthma in Children

**DOI:** 10.3390/biology12010133

**Published:** 2023-01-13

**Authors:** Serena Di Vincenzo, Giuliana Ferrante, Maria Ferraro, Caterina Cascio, Velia Malizia, Amelia Licari, Stefania La Grutta, Elisabetta Pace

**Affiliations:** 1Institute of Translational Pharmacology (IFT), National Research Council of Italy (CNR), 90146 Palermo, Italy; 2Pediatric Unit, Department of Surgical Science, Dentistry, Gynecology and Pediatrics, University of Verona, 37126 Verona, Italy; 3Pediatric Unit, Department of Clinical, Surgical, Diagnostic and Pediatric Sciences, University of Pavia, 27100 Pavia, Italy; 4Pediatric Clinic, Fondazione IRCCS Policlinico San Matteo, 27100 Pavia, Italy

**Keywords:** oxidative stress, lifestyle, asthma, children

## Abstract

**Simple Summary:**

This review is focused on the role of oxidative stress in childhood asthma. The term “oxidative stress" indicates an imbalance between production of radical oxygen species and antioxidant defense systems. Cigarette smoke, allergens, viruses, and other environmental contaminants, as well as a detrimental lifestyle, elevate airway oxidative stress. Uncontrolled oxidative stress within airways amplifies inflammatory responses and tissue damage and alters immune responses, thus increasing the risk of asthma onset and asthma progression in children. Early lifestyle and dietary interventions and use of new antioxidant therapies to restore oxidant/antioxidant balance are all valuable strategies to preserve airway homeostasis.

**Abstract:**

Exposure to cigarette smoke, allergens, viruses, and other environmental contaminants, as well as a detrimental lifestyle, are the main factors supporting elevated levels of airway oxidative stress. Elevated oxidative stress results from an imbalance in reactive oxygen species (ROS) production and efficiency in antioxidant defense systems. Uncontrolled increased oxidative stress amplifies inflammatory processes and tissue damage and alters innate and adaptive immunity, thus compromising airway homeostasis. Oxidative stress events reduce responsiveness to corticosteroids. These events can increase risk of asthma into adolescence and prompt evolution of asthma toward its most severe forms. Development of new therapies aimed to restore oxidant/antioxidant balance and active interventions aimed to improve physical activity and quality/quantity of food are all necessary strategies to prevent asthma onset and avoid in asthmatics evolution toward severe forms of the disease.

## 1. Introduction

Uncontrolled increased oxidative stress sustains pathophysiology of different diseases [1,2,3]. Asthma is a chronic inflammatory disease characterized by reversible obstruction of the airways. In asthmatic patients, reactive oxygen species (ROS) production and consequent oxidative stress levels are elevated compared to healthy subjects [4,5,6,7,8,9,10]. A vicious cycle of events supports the increased level of oxidative stress within the airways of asthmatics: external insults (allergens, pollution, smoke, etc.) trigger oxidative stress events that, in turn, promote pro-inflammatory events, including recruitment of activated inflammatory cells (eosinophils, neutrophils, and macrophages) that further increase oxidative stress since these cells are the primary source of ROS [11]. Uncontrolled oxidative stress amplifies airway structural damage, accelerates remodeling processes, and alters the response to corticosteroids [12,13]. All these events in children with asthma favor evolution of asthma toward the most severe asthma forms. This review offers a comprehensive overview of the molecular mechanisms that lead to increased oxidative stress, describes the impact of oxidative stress on pediatric asthma, and suggests some lifestyle interventions to counteract oxidative stress.

## 2. Mechanism of Oxidative Stress and Asthma Correlation

The expression “oxidative stress” was first introduced in 1985 by Sies [14], who defined it as an imbalance between production of radical oxygen species and antioxidant defense systems. Several mechanisms can determine oxidative stress: overproduction of ROS and reactive nitrogen species (RNS) in chronic inflammation; deficiency in intrinsic (e.g., glutathione) or extrinsic (e.g., vitamins and natural antioxidants in diet) antioxidant molecules; decreased activity or dysfunction of antioxidant enzymes; and over-activity of pro-oxidative enzymes [11]. There are two types of ROS sources: endogenous and exogenous. Endogenous ROS are produced from molecular oxygen due to normal cellular metabolism, enzymatic oxidation, by phagocytic cells, or during inflammatory phenomena. Exogenous sources of ROS are some drugs (e.g., chemotherapeutic agents), radiation, metal ion transitions, damage from pulmonary circulatory alterations (ischemia–reperfusion), and environmental pollution with inorganic particles, such as asbestos, quartz, silica, exhaust gases, and cigarette smoke. Both sources of ROS contribute to the inflammatory conditions worsening, causing an accumulation of oxidized species [11]. In physiological conditions, the cell maintains a reducing state inside thanks to enzymes and molecules that counterbalance ROS production, which is an antioxidant system. Two classes of antioxidants are known: non-enzymatic antioxidants, such as vitamins A, C, and E, flavonoids and procyanidins, β-carotene, and glutathione; and enzymatic antioxidants, such as superoxide dismutase (SOD), catalase, heme oxygenase-1, peroxiredoxins, thioredoxins, and glutaredoxins.

In asthma, airway inflammation is an important source of ROS. Activated inflammatory cells (eosinophils, neutrophils, and macrophages) are the primary source of ROS. If not balanced by the functional antioxidant systems, an excessive increase in ROS levels leads to oxidative stress that causes structural and functional damage to the various biomolecules (lipids, proteins, and DNA), altering biological processes [15]. In this regard, oxidative stress causes oxidation reactions involving sugars or their degradation products, leading to formation of advanced glycation end products (AGEs) [16]. Activation of the receptor for AGE (RAGE) is involved in the inflammatory process associated with the pathophysiology of asthma [17] and seems to be a potential biomarker and therapeutic target [18]. Furthermore, oxidative stress activates transcription factors (NF-κB, AP-1), which regulate expression of genes for pro-inflammatory cytokines, enzymes, and adhesion molecules that induce an inflammatory response [19,20,21]. Therefore, there is a dynamic relationship between increased ROS levels and the inflammatory processes characteristic of asthma. Another transcription factor involved in oxidative stress mechanisms is nuclear factor erythroid 2-related factor 2 (Nrf2), a molecule that binds the antioxidant responsive element (ARE) promoters present in genes encoding antioxidant enzymes [22]. Extracellular superoxide dismutase (EC-SOD) is thought to play a role in the mechanism of asthma pathology [23]. EC-SOD is an antioxidant enzyme present in the matrix around the airways [24] whose expression is regulated by Nrf2. Use of EC-SOD mimics in animal models with asthma has been shown to decrease airway inflammation [25]. Therefore, increased EC-SOD function, which primarily decreases oxidative stress, could reduce asthma symptoms [26]. Glutathione (GSH) is another antioxidant enzyme controlled by Nrf2. This enzyme is expressed in epithelial lining fluid (ELF) and within airway cells [27,28]. GSH oxidizes to reduce hydroperoxides in water to form glutathione disulfide (GSSG). Low GSH levels were found in exhaled breath [29] and ELF [30] of asthmatic children. Corticosteroid treatment normalizes the GSH to GSSG ratio of the airways in children with mild to moderate asthma [29], but this is not the case in children with severe asthma, who maintain marked GSH depletion with increased lipid peroxidation in the airways [30]. This imbalance promotes histone acetyltransferase (HAT) activity and inhibits histone deacetylase (HDAC) activity in airway cells, leading to increased release of proinflammatory cytokines, such as IL-8 [31]. This explains why GSH supplementation in ovalbumin-sensitized mice leads to restoring GSH/GSSG balance in airway tissue and reduces airway dysfunction [32].

## 3. Impact of Oxidative Stress on Airway Homeostasis

The airway epithelium is the first barrier protecting the airways from environmental insults, but it is also an essential component of innate immune responses. Environmental insults (cigarette smoke, infections, pollutants) activate oxidative stress not only by inducing ROS but also by suppressing the antioxidant response [11]. Epithelial cells respond to these external insults by activating innate responses (by Toll-like receptors (TLRs)), releasing defensins and mucus, coordinating local inflammation and the immune response through release of cytokines and chemokines, activating antioxidant responses, and maintaining overall tissue homeostasis [12]. Exposure of the airway epithelium to environmental insults in early life may lead to permanent changes in structure and function that underlie development of asthma [33,34]. At birth, the epithelial barrier is formed but the immune system is immature, so, in the first years of life, children are more vulnerable to external insults [35]. Airway remodeling following exposure to environmental insults was observed in preschool children before clinical diagnosis of asthma [36]. Prenatal and early postnatal exposure to passive cigarette smoke induces adverse effects on both immune system and lung development, thus increasing risk of asthma onset [37]. Oxidative stress induced by exposure to cigarette smoke of bronchial epithelial cells increases TLR4 expression and activation by bronchial epithelial cells [38], leading to increased pro-inflammatory responses. It also alters barrier function acting on the organization of the cytoskeleton and remodeling of the airways [39].

Corticosteroids are widely used in asthma therapy. Patients with severe asthma require high doses of corticosteroids, but reduced efficacy results in persistent airway inflammation, airflow obstruction, and more frequent exacerbations [40]. Recent studies have highlighted the role of ROS in inducing or worsening glucocorticoid insensitivity in severe and difficult-to-treat asthma, impairing glucocorticoid receptor (GR) both structurally and functionally [4,5,6,41]. ROS overproduction leads to activation of p38 mitogen-activated protein kinases (p38MAPK) [5,6,41,42]. This class of molecules mediates phosphorylation and inactivation of GR and subsequent inactivation of histone deacetylase 2 (HDAC2), a crucial co-repressor of many pro-inflammatory transcriptional factors [4,6,41], such as nuclear factor kappa-light-chain-enhancer of activated B cells (NF-κB) and activator protein-1 (AP-1). Studies have also shown the capability of ROS excess to activate phosphoinositide 3-kinase δ (PI3Kδ), a pro-inflammatory enzyme involved in T cell signaling, mast-cell activity, and HDAC2 inhibition [41,42]. Finally, chronic ROS hyperproduction has also been linked to alteration in β2-adrenergic receptor signaling [4]. In children with difficult-to-treat asthma, high oxidation of cysteine is observed [43]; also, passive smoking impairs HDAC2 function [44]. Exposure of bronchial epithelial cells to cigarette smoke decreased HDAC3 and HDAC2 activity/expression and nuclear translocation of the glucocorticoid receptor (GR), contributing to steroid resistance [45,46]. Taken together, these mechanisms could explain the reduced responsiveness to corticosteroids. Various studies show that adding drugs with antioxidant actions enhances the effects of steroids in conditions of high oxidative stress [13,45,46].

## 4. Antioxidant/Oxidant Balance and Immune Responses

Together with the airway epithelium, immune cells (alveolar macrophages, dendritic cells, T cells, innate lymphoid cells, and granulocytes) present in the submucosa provide a response to external agents.

Oxidative stress is considered to be a component of asthma pathophysiology. High levels of ROS activate expression of pro-inflammatory genes, with a consequent increase in adhesion molecules and release of pro-inflammatory mediators, cytokines, and chemokines [47]. Furthermore, the permeability of endothelial cells and expression of adhesion molecules are altered, leading to recall of inflammatory cells and a more significant adhesive interaction between these cells and the endothelium [48,49]. The most common immune cells in the airways are macrophages. These cells also generate ROS and reactive nitrogen species (RNS) to protect against invading pathogens [50,51]. It has been shown that children with severe asthma have an unbalanced GSH/GSSG ratio toward increased GSSG levels in epithelial lining fluid (ELF) and at the intracellular level of alveolar macrophages [52]. This leads to increased macrophage apoptosis for caspase activation and Poly (ADP-ribose) polymerase (PARP) degradation [53] and reduced phagocytosis of pathogens [54] through Nrf2 inhibition [55]. This has also been associated with decreased HDAC activities [52]. Accumulation of oxidant species can lead to oxidation of proteins with fundamental biological functions, questioning their activity. A possible oxidation target can be Surfactant Protein A (SP-A), an innate immunity molecule that opsonizes some pathogens and facilitates phagocytosis by alveolar macrophages, induces expression of inflammatory mediators, and leads to maturation of dendritic cells [56]. Some in vitro studies demonstrate that oxidation of SP-A due to ozone reduces phagocytosis by alveolar macrophages [57,58,59].

Neutrophils and eosinophils recruited in the airways are an additional source of ROS. Neutrophils express enzymes nicotinamide adenine dinucleotide phosphate (NADPH) oxidase and myeloperoxidase. When activated thanks to these enzymes, these cells produce superoxide anion and hypochlorous acid, which are used in the response against pathogens [60]. However, if the neutrophil activation occurs by air pollutants, the transition metals present within the particulate matter (PM) can catalyze redox cycles and damage the biological systems [61,62]. Activation of eosinophils following exposure to viruses or allergens also produces ROS thanks to release of eosinophilic peroxidase (EPO), which brominates amino acid tyrosine [63].

Oxidative stress unbalance can also modify adaptive immunity responses. It influences antigen-presenting cells (APC) and T cells responses, shifting the Th1/Th2 balance toward Th2 responses [64,65,66,67,68]. Some studies demonstrate that expression of co-stimulatory molecules and release of IL-12 by DCs stimulated with lipopolysaccharides (LPS) is reduced in conditions of low levels of GSH [64] and decreases production of IFNγ in mice without any decrease in IL-4 production [65]. Furthermore, it has been shown that oxidative stress caused by air pollution exposure could inhibit activation of dendritic cells, a particular type of APC, induced by TLRs. This would decrease Th1 response with reduced IFN-γ production in T lymphocytes [66,67]. In ovalbumin-sensitized mice, treatment with a GSH mimic reverts the T lymphocyte response toward the Th1 response [62,68]. Taken together, these findings suggest that deficiency in Th1 responses and the reduced levels of IFNγ due to increased oxidative stress in asthmatics could affect the efficiency of anti-viral responses, thus favoring occurrence of asthma exacerbations and progression. Figure 1 shows a summary of proposed mechanisms that contribute to oxidant/antioxidant imbalance in asthma pathophysiology.

## 5. Environmental Pollution (Prenatal, Natal, and Postnatal Exposure) and Oxidative Stress

Environmental pollution has been widely recognized as a major risk factor for human health. During recent decades, research has highlighted the deleterious effects of environmental pollutants on respiratory health of adults and children. In particular, air pollution is currently receiving great attention, and the relationship between exposure in childhood and development or exacerbation of adverse respiratory outcomes, including asthma, is becoming more evident [69,70,71,72]. Indeed, several air pollutants are known to impair lung development and function, indirectly (prenatal exposure) and directly (postnatal exposure). Oxidative stress and DNA damage are plausible mechanisms explaining how air pollution can affect respiratory health [73]. Indeed, outdoor and indoor pollutants can cause lung injury due to oxidative stress by acting directly on ROS production, or indirectly by inducing inflammation. ROS are a normal product of cell metabolism and are able to react with cellular components, such as DNA and membrane lipids, contributing to epithelial cell inflammation, airway hyperreactivity, tight junction barrier permeability, and lung injury [74,75].

Confirming previous evidence in animal models demonstrating a role of air pollution exposure during pregnancy in increasing risk of asthma in offspring [76,77], findings from a recent cohort study showed that prenatal exposures, including air pollution and maternal smoking, which are thought to contribute to oxidative imbalance in utero, are associated with increased risk of asthma into adolescence [78]. On the other hand, the authors did not observe any beneficial effects of prenatal administration of antioxidants such as vitamin C and beta-carotene.

However, the available evidence has been mainly focused on the effects promoted by postnatal exposures. Patel et al. [79] attempted to evaluate associations between ambient diesel exhaust particle exposures and biomarkers of airway inflammation and oxidative stress in exhaled breath condensate from adolescents with and without asthma. Among all the participants, regardless of asthma status, increases in 1- to 5-day averages of black carbon were associated with decreased pH, suggesting increased airway inflammation [80] and increased 8-isoprostane, indicating increased oxidative stress [81]. Noteworthily, increases in 1- to 5-day averages of nitrogen dioxide were also associated with increased 8-isoprostane [79].

A study involving schoolchildren with persistent asthma reported a significant positive association between fractional exhaled nitric oxide (FeNO) levels and particulate matter <2.5 in diameter (PM_2.5_) oxidative potential rather than PM_2.5_ mass, suggesting that research in asthma and PM exposure would benefit from assessments of PM oxidative potential [82]. Indeed, similar findings were more recently reported by Yang et al. [83], showing that respiratory health of children enrolled in the Prevention and Incidence of Asthma and Mite Allergy (PIAMA) prospective birth cohort was more strongly associated with PM_2.5_ oxidative potential than with PM_2.5_ mass. In particular, PM_2.5_ oxidative potential was associated with increased risks of asthma up to 14 years and decreased lung function at the age of 12 years [83]. A study by He et al. [84] on children with mild to moderate asthma confirmed that increases in daily personal exposure to PM_2.5_ oxidative potential were significantly associated with increased airway impedance, decreased lung function assessed by spirometry, and worsened scores of asthma symptoms. Among the PM_2.5_ constituents, indoor organic matter resulted as a major contributor to PM_2.5_ oxidative potential [84].

The relationship between volatile organic compounds (VOCs) exposure and oxidative stress in childhood asthma was evaluated in a recent case-control study by Kuang et al. [85]. Levels of urinary 8-hydroxy-2′-deox-yguanosine (8-OHdG, a biomarker of oxidative DNA damage) and VOC metabolites in children with asthma were significantly higher than those observed in healthy children. Most of the VOC metabolites were significantly and positively associated with trans-3′-hydroxycotinine (OH-Cot, a biomarker of passive smoking), indicating second-hand smoke as a relevant source of VOCs exposure in children. Significant dose–response relationships between most VOC metabolites and 8-OHdG were observed. Moreover, each unit increase in 8-OHdG level was associated with an increased risk of asthma in children, indicating that VOCs exposure may significantly impact childhood respiratory health through increased oxidative stress and DNA damage [85].

Moreover, the effect of polycyclic aromatic hydrocarbons (PAHs) on asthma in children has been suggested to be mediated by oxidative stress as exposure was associated with 8-OHdG in urine in a case-control study by Wang et al. [86], suggesting that exposure to PAHs may enhance oxidative stress and induce asthma. In particular, through mediation analysis, the authors estimated that 35% of the effect of PAHs exposure on asthma is mediated by 8-OHdG levels [86]. Recently, Cilluffo et al. [71], in a single-center prospective study involving 50 children with persistent mild–moderate asthma aged 6–11 years, showed that PAHs exposures have significant indirect (symptom-mediated) effects on lung function, emphasizing the role of PAHs-induced respiratory morbidity in decreasing lung function in children with asthma.

Combined, these findings suggest that both prenatal and postnatal exposure to air pollutants may increase airway inflammation and/or oxidative stress, providing mechanistic support for epidemiological evidence of associations with asthma morbidity in pediatric age. Along with the public health strategy of reducing children’s exposure to air pollution, reducing pollution-induced oxidative stress and consequent lung damage through administration of antioxidant compounds may be considered as it would be a low-cost intervention that children could easily undertake. Evidence supports the role of vitamin C in protecting the airways from oxidative injury induced by exposure to air pollution. In a previous study by Romieu et al. on children with asthma exposed to high levels of ozone, it was found that those with a Glutathione S-transferase (GST)M1 null genotype had a greater decline in lung function and received greater benefit from antioxidant administration [87]. However, further evidence is required to confirm whether antioxidant supplementation could be biologically meaningful in children with asthma, especially in those with a genetic susceptibility to oxidative stress. Research should also address proper timing of antioxidants administration to achieve an effective prevention effect.

Finally, there is increasing evidence that the detrimental effects of air pollution on respiratory health may be influenced by variations in antioxidant genes. Future studies in this research field are warranted to understand better the pathogenetic mechanisms underlying health outcomes, including assessing the potential role of epigenetics in explaining the interactions of antioxidant genes with air pollution. Furthermore, identifying the critical time windows of exposure will be crucial for developing effective preventive strategies to protect this vulnerable population against the adverse effects of air pollution [88].

## 6. The Role of Reactive Oxygen and Nitrogen Species in Childhood Asthma

In recent years, the correlation between asthma, especially severe asthma, and redox imbalance in subjects of all ages has been a matter of growing interest in the literature in an attempt to characterize better asthma subgroups (the so-called “endotypes” and “phenotypes”) and to design personalized, steroid-sparing therapies complementary to the traditional ones [4,5,6,7,8,9,10].

As previously described in this review, redox imbalance in asthma is characterized by ROS overproduction and quantitative or qualitative loss of antioxidant mechanism [4,5,10]. The increase in the endogenous production of ROS and reactive nitrogen species (RNS) in asthmatics, due to insult of both structural cells (epithelium, fibroblasts, smooth muscle cells, etc.) and cells of the immune system (macrophages, neutrophils, eosinophils, etc.), seems to be directly linked to severity of asthma [4,7,10], even in the case of apparently good clinical control [89].

Studies have shown that ROS excess in airway epithelium increases asthma inflammation by releasing tumor necrosis factor (TNF) and IL-1, IL-6, and IL-8, especially with regard to severe asthma, asthma attacks, and asthma in obese subjects [4,8]. This may be due to direct oxidative damage and indirect alteration in redox signaling, globally promoting inflammation and hindering free radical detoxification [4,7]. In turn, both eosinophilic and neutrophilic inflammatory infiltrates are significant producers of ROS with their oxidative burst [4], thus promoting an unstoppable futile circle of inflammation and oxidation, which magnifies the processes of airway remodeling and hyperresponsiveness. This is particularly true for severe asthmatics as their eosinophils and neutrophils show higher levels of nicotinamide adenine dinucleotide phosphate (NADPH) oxidase 2 [4,90], one of the most relevant enzymes responsible for O2^−^ production. Furthermore, ROS excess in asthmatics leads to loss of proper intracellular redox sensing and dysfunction of subcellular organelles, mainly mitochondria [4,7,8], which are thought to be involved in airway smooth muscle cells contractility, airway thickening, and air-pollution-mediated asthma attacks [4].

Exogenous sources of ROS are critical in asthma pathogenesis: they can induce asthma attacks and promote local inflammation [4,7]. It is known that aeroallergens can act as pathogen- or danger-associated-molecular-patterns (PAMPS and DAMPS), triggering specific pattern recognition receptors (PRRs), which promote T-helper 2 (Th2) or Th17 inflammation [4]. PM2.5 and 0.1 can activate inflammatory pathways as well, interacting with redox sensors in airway epithelium and resident macrophages and increasing neutrophilic infiltration through IL-17 induction [5]. Moreover, they lead to ROS production through action of cytochrome p450, which metabolizes the polycyclic aromatic hydrocarbons in them [4]. Ozone has proven to be an inductor of type 2 high bronchial inflammation through IL-33 release in animal models [4,5], capable of promoting airway hyperresponsiveness [5]. Interestingly, it has been shown that aeroallergens possess, among their non-antigenic components, proteins with intrinsic NOX activity, which makes them capable of directly inducing production of free radicals [5]. Finally, the thousands of toxic chemicals in passive and active smoke increase mitochondrial ROS production and appear to be involved in glucocorticoid resistance [5,90]. Noteworthily, they can also induce type 2 high inflammation through release of alarmins, such as thymic stromal lymphopoietin (TSLP) [4,5].

Through epithelial damage and inflammatory cell recruitment, both endogenous and exogenous ROS determine release of growth factors, such as epidermal growth factor (EGF) and transforming growth factor-beta (TGF-β), finally leading to airway remodeling [4,5]. In vitro tests have also shown a potential role for ROS in induction of adaptive immune responses (through T lymphocytes maturation and activation) and airway hyperresponsiveness (promoting proliferation and contraction of smooth muscle cells) [4,5,91].

Moreover, ROS direct interaction with specific surface receptors on airway epithelial cells is thought to promote allergic sensitization [4], highlighting a potentially early role of redox imbalance in asthma pathogenesis.

All this considered, ROS and RNS derivatives have been proposed as asthma biomarkers, especially regarding severe, neutrophilic asthma [4,5,6,89]. Although ROS are too reactive, unstable, and compartmentalized to be directly measured, in recent years, there has been a growing increase in studies carried out by multidisciplinary research groups for development of electrochemical sensors that enable measurement of some more stable ROS in cell line culture media with the aim of being able to apply them directly on patients [92,93]. Furthermore, many secondary products of their excess have been proposed as indirect redox markers in many different biological samples, such as urine, blood, broncho-alveolar lavage (BAL), induced sputum, or exhaled breath condensate [4,5,7,10,59,89,90,94]. These biomarkers are a promising, emerging field of research as, in some studies, they happened to be increased in case of severe asthma, asthma attacks, or steroid-resistant asthma [4,5,89,90]. Moreover, in some studies, they also seem to correlate with poor asthma control and worse lung function [4,9]. Therefore, large, high-quality prospective studies may lead to their future use in severe asthma as endotyping tools, asthma control tools, or steroid-resistance markers [8]. Since some of these redox derivatives seem to correlate with more well-known markers, such as sputum eosinophils [4] or FeNO [5,10,90], comparing old and new markers can enable definition of clear, meaningful cut-offs to be used for clinical practice. Table 1 summarizes the most promising indirect redox markers described in the literature, the most relevant of which are serum malondialdehyde, urinary and BAL bromotyrosine, and serum and urinary isoprostanes as they seem to correlate with asthma severity [4] and poor asthma control [7].

Finally, given their role in pathogenesis of asthma and their involvement in glucocorticoid insensitivity, ROS and RNS could become potential therapeutic asthma targets as add-on steroid-sparing treatments. However, doses, routes, and administration intervals are yet to be defined [5]. Corticosteroids and antioxidants can work synergistically: the former reduces the airway pro-oxidative state [4] and the latter restores optimal glucocorticoid sensitivity [5,42] to reduce bronchial hyperresponsiveness, increase asthma control, and preserve lung function. Redox balancing therapies may improve endogenous antioxidant mechanisms or reduce ROS and RNS concentration. The options currently under investigation, mostly in preclinical studies, include dietary introduction of vitamins and polyphenols [4,6,7,10]; selective NOX inhibitors [4]; administration of N-acetylcysteine, proven to reduce air-pollution-mediated bronchial hyperresponsiveness and reliever use [4,7,46]; Nrf2 agonists such as sulforaphane [4,5], leading to activation of anti-inflammatory signaling; superoxide dismutase mimicking molecules; coenzyme Q, a component of the mitochondrial electron transport chain capable of decreasing mitochondrial redox dysfunction [4,7,8]; inhaled p38MAPK inhibitors [5,41,42]; and selective and not-selective PI3Kδ inhibitors [41], some of them capable of simultaneously activating HDAC2, thus inducing transcription of anti-inflammatory genes and restoration of proper steroid sensitivity [41]. Interestingly, macrolides can also be used as PI3Kδ inhibitors more distally in the molecular pathway thanks to their anti-inflammatory properties [41,42].

All these therapies have not yet provided encouraging, consistent results due to the lack of identification of subjects with oxidant/antioxidant unbalance and standardized therapeutic schemes, small samples sizes, complexity of redox signaling (which is strictly compartmentalized in space and time), and reduced in vivo stability of some molecules [4,5,8,99]. Therefore, novel drug delivery systems are currently under study to ensure strategical use of nanotechnology organelle-targeted therapies capable of reaching specific subcellular sites [99]. Using such nebulized or inhaled nanoparticles, liposomes, and microemulsion, with peculiar pharmacokinetic properties in terms of diffusion, solubility, and dimensions, may lead to maximum tailoring of therapies [99], thus reducing side effects and, finally, increasing compliance. Nano- and micro-structured lipid carriers can be considered a promising strategy to improve corticosteroid-mediated effects in a cellular model associated with high oxidative stress and corticosteroid resistance [100,101,102].

Further studies are required to prove the clinical utility and availability of ROS and RNS products and antagonists in severe asthma as clinical biomarkers and future innovative therapies, finally returning from threat to breath.

## 7. Lifestyle Interventions to Counteract Oxidative Stress in Childhood Asthma: The Role of Diet and Physical Activity

Unbalanced dietary intake has been increasingly recognized as an important modifiable risk factor for asthma [103]. Accordingly, dietary interventions have been proposed as an approach to reduce asthma incidence or morbidity given the proven adverse effect of a pro-inflammatory diet on asthma burden [103]. In particular, there is evidence of an association between low dietary intake of antioxidants and increased asthma prevalence [104] and that specific antioxidant supplements might improve disease control or lung function in asthmatic patients [105].

Vitamin C is the most recognized antioxidant and has thus been hypothesized as a complementary treatment for asthma [106]. Its intake has been suggested to counteract oxidative damage in the lungs by blocking activation of the NF-κB pathway by inhibiting TNF-α formation and altering the arachidonic acid pathway. Indeed, vitamin C deficiency has been recently associated with severe asthma and airway obstruction in children [95]. Nonetheless, most of the available studies assessing the effect of vitamin C supplementation on asthma symptoms and pulmonary function reported inconsistent results, suggesting that more research is needed to evaluate its role as a supportive tool in asthma management [96]. Noteworthily, efficacy of vitamin C in asthma management may rely on antioxidant interdependency, resulting in improved results when administered in whole foods, such as fruits and vegetables. According to the systematic review by Hosseini et al. [97], most of the studies included reported beneficial associations between fruit and vegetable consumption and risk of asthma and pulmonary function. In addition, meta-analyses in studies of both adults and children showed inverse associations between fruit intake and risk of prevalent wheeze and asthma severity (*p* < 0.05) [97]. More recently, Mendes et al. [98] demonstrated that consumption of a greater variety of vegetables was associated with lower risk of airway inflammation (OR = 0.38; 95% CI 0.16, 0.88) and self-reported asthma (OR = 0.67; 95% CI 0.47, 0.95) in a general population of schoolchildren depending on amount of vegetable intake. These results add further evidence that consuming healthy and plant-based diets may contribute to improving respiratory health in children [98].

Selenium is a micronutrient involved in the antioxidant defense mechanisms as a cofactor of glutathione peroxidase [107]. Selenium seems to promote Th1 cells differentiation by inducing Th1 cytokines expression and/or inhibiting Th2 cytokines secretion [108]. Along with selenium, zinc is one of the most important micronutrients implicated in maintaining oxidant–antioxidant balance, mainly by inhibiting pro-oxidant enzymes, such as NADPH oxidase and inducible nitric oxide synthase, and activating antioxidant enzymes, such as glutathione-related enzymes, catalase, and superoxide dismutase [109]. Furthermore, zinc has been suggested to possess immunomodulatory effects by restoring the impaired balance between Th1 and Th2 cells [110]. Serum selenium and zinc were found to be lower in children with asthma with respect to controls (*p* = 0.033 and *p* = 0.008, respectively), selenium levels being significantly lower in those with moderate/severe compared to mild disease (*p* = 0.001) [111]. More recently, Siripornpanich et al. [95] found a correlation between zinc levels and spirometry parameters, such as forced expiratory volume in 1 s (FEV_1_) and forced expiratory volume in 1 s, to forced vital capacity ratio (FEV_1_/FVC) in children with asthma. Interestingly, a randomized, placebo-controlled study involving children with persistent asthma on inhaled corticosteroids treatment previously demonstrated that eight-week zinc supplementation improved both symptoms, such as wheezing, cough, and dyspnea, and spirometry parameters, such as FVC, FEV_1_, and FEV_1_/FVC [112]. Combined, these findings suggest that selenium and zinc supplementation could contribute to maintaining the balance of pro- and antioxidative as well as pro- and anti-inflammatory responses, thereby improving asthma symptoms and pulmonary function.

Physical activity (PA) significantly contributes to a healthy lifestyle and better health-related quality of life [113,114]. Evidence suggests that PA is beneficial in combating chronic respiratory diseases such as asthma [115]. A pilot study shows that lower PA is associated with higher fat mass and severity of asthma [116]. Indeed, it has been shown to improve asthma symptoms, pulmonary function, and quality of life, as well as to reduce airway inflammation [117,118]. Furthermore, there is evidence that PA can modulate immune system functioning, reducing systemic inflammation and being protective against overweight and obesity [119]. Previous studies evaluated the effects of PA on childhood asthma to define its potential benefits. Onur et al. [120] evaluated blood levels of malondialdehyde (MDA) and total nitric oxide (NO), glutathione peroxidase (GSH-Px), and superoxide dismutase (SOD) enzyme activities in a case-control study. The authors found that pharmacological treatment and exercise programs together significantly decreased levels of oxidant stress markers and increased antioxidant enzyme activity compared to the pharmacological treatment alone [120]. More recently, salivary levels of MDA were found to significantly decrease after exercise compared to baseline (*p* < 0.05) in obese and non-obese children with asthma, suggesting a potential benefit of PA on antioxidant status [121]. Such evidence has been further supported by a systematic review including studies on adults and children demonstrating that physical training may reduce airway inflammation in asthmatics. However, markers of airway inflammation used, physical training interventions (type, intensity, duration, frequency), and control interventions were heterogeneous across studies, thereby limiting the strength of conclusions [122]. This highlights the need for more research in this area to assess whether regular PA may be a potential non-pharmacologic treatment in children with asthma.

Evidence suggests that children with asthma tend to be less physically active than their healthy peers [123,124]. Limiting PA, sedentary behaviors may lead to an increase in adiposity, and this could be detrimental, especially in Westernized countries, where both asthma and obesity are common. Obesity is an inflammatory condition that may contribute to increase asthma prevalence and may worsen asthma control and severity, being associated with more frequent exacerbations, reduced response to asthma medications, and decreased quality of life [125]. Lifestyle changes are generally the first approach for addressing overweight and obesity in children. A strategy combining both dietary and exercise interventions theoretically should be the most effective. Interestingly, a recent study investigated the effects of an 8-week physical training program and/or a dietary intervention in non-obese adults with asthma [126]. The dietary intervention consisted of a high-protein and low-glycemic-index diet, including high amounts of vegetables, fruits, and fish. Although a significant improvement in asthma control and quality of life was observed in the group receiving the combined intervention compared to the control, there was no significant change in airway inflammation. Moreover, it should be pointed out that the combined intervention group also lost a significant amount of weight, so the findings might be due to weight loss. However, no similar studies have been performed on children so far. This research area requires more investigation in children with asthma.

In summary, changes in diet, including increasing fruit and vegetable intake, along with regular PA may improve antioxidant status and respiratory health. Proposed pharmacological and lifestyle interventions that might prevent oxidant/antioxidant imbalance in asthma are described in Figure 2.

## 8. Conclusions

Oxidant/antioxidant imbalance is a crucial phenomenon that affects airway homeostasis, modulating inflammation and innate and adaptive immunity. All these events exert a fundamental role in asthma physiopathology. The efficacy of primary, secondary, and tertiary prevention interventions that maintain oxidant/antioxidant balance are useful strategies to limit asthma onset or progression (Figure 3). Thus, there is an urgent need to improve diagnostic tools to identify subjects or patients with uncontrolled oxidative stress within the airways. Early identification of oxidative stress may orientate lifestyle and dietary interventions and use of new antioxidant therapies to restore oxidant/antioxidant balance (Table 2).

## Figures and Tables

**Figure 1 biology-12-00133-f001:**
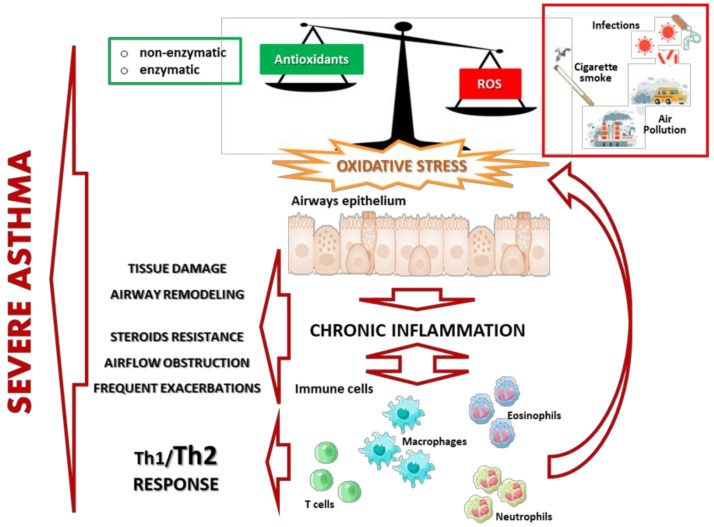
Summary of proposed mechanisms that contribute to oxidant/antioxidant imbalance in asthma pathophysiology.

**Figure 2 biology-12-00133-f002:**
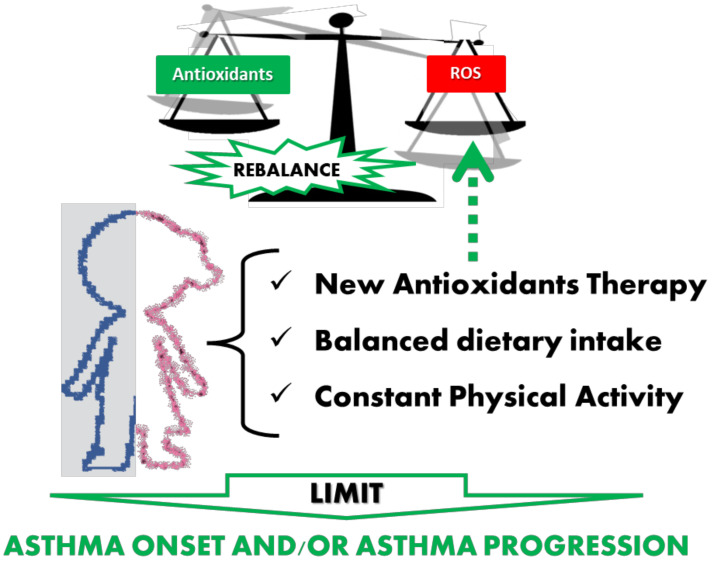
Proposed pharmacological and lifestyle interventions that might prevent oxidant/antioxidant imbalance in asthma.

**Figure 3 biology-12-00133-f003:**
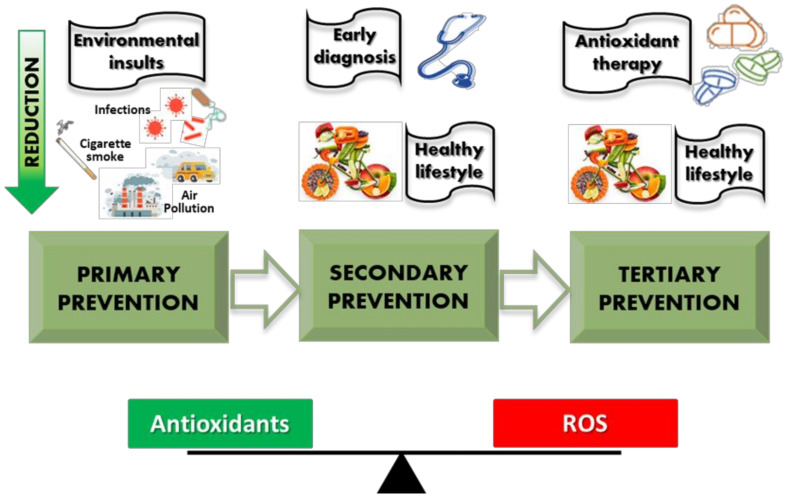
Primary, secondary, and tertiary preventive interventions to reduce oxidative stress.

**Table 1 biology-12-00133-t001:** Main indirect redox markers of asthma and severe asthma currently described in the literature.

Products of lipid oxidation [15,30,74,75] Phospholipid oxidation [91]	Malondialdehyde (correlate with asthma severity) Nitro tyrosine 4-hydroxynonenal 8-isoprostanes Urinary 8-iso-PGF2α (correlate with asthma severity, asthma attacks and airway hyperreactivity) Oxidized phosphatidylcholines
Products of oxidative DNA damage [15,73,74,75,85]	Urinary 8-oxo-7,8-dihydro-2-deoxyguanosine
Products of protein oxidation [5,15,57,58,59]	Asymmetric dimethylarginine Protein carbonyls
Products of eosinophils/neutrophils peroxidase [29,30]	Bromotyrosine Chlorotyrosine
Total antioxidant capacity indicators (FRAP) [95,96,97,98]	Uric acid Ascorbic acid Total bilirubin
Antioxidant imbalance (both quantitative and/or qualitative) [27,30,32,52,54]	GSH/GSSG Paraoxonase Glutathione peroxidase activity Catalase activity
EBC markers [4,7,79]	pH (H^−^ concentration) NO_2_, NO_3_ H_2_O_2_ 8-isoprostanes MitDNA

PGF2 α: prostaglandin F2 α; FRAP: ferric reducing ability of plasma; GSH: glutathione; GSSG: glutathione disulfide; EBC: exhaled breath condensate; NO_2_: nitrites; NO_3_: nitrates; H_2_O_2_; hydrogen peroxide; MitDNA: mitochondrial DNA.

**Table 2 biology-12-00133-t002:** Areas of research needs in children with asthma.

Areas of Research Needs in Children with Asthma
Identification of the critical time windows of exposure for developing effective preventive strategies to protect children with asthma against the adverse effects of air pollution. Identification of clinical biomarkers of oxidative stress in children with asthma at all severity levels. Investigation of the effects of dietary intervention on airway inflammatory pathways in children with asthma.

## Data Availability

Not applicable.

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
