# Peer review of "Oxidative Stress, Environmental Pollution, and Lifestyle as Determinants of Asthma in Children"

_biology, 2023, doi:10.3390/biology12010133_

Round 1

Reviewer 1 Report

Introduction: 

Please provide literature/references to support the following statements:

 “In asthmatic patients, reactive oxygen species (ROS) production and consequent oxidative stress levels are elevated compared to healthy subjects. A vicious cycle of events supports the increased level of oxidative stress within the airways of asthmatics: external insults (allergens, pollution, smoke, etc) trigger oxidative stress events that, in turn, promote pro-inflammatory events, including the recruitment of activated inflammatory cells (eosinophils, neutrophils, and macrophages) that further increased oxidative stress since these cells are the primary source of ROS”

Main Text

Page 3, the following sentence should be modified to specify which insults – for example, not all insults are predicted to activate toll-like receptors 

“The epithelial cells respond to external insults by activating innate responses (by Toll-like receptors (TLRs)), releasing defensins and mucus, coordinating local inflammation and the immune response through the release of cytokines and chemokines, activating antioxidant responses, and maintaining overall tissue homeostasis. “

Page 3, The following statement is not entirely correct.  The epithelial barrier is formed; however, the immune system is immature.  Please correct this statement accordingly: “At birth, the epithelial barrier is not fully formed, so in the first years of life, children are more vulnerable to external insults [23]”

Page 3, lines 112-113 - The authors suggest that cigarette smoke is one of the most significant sources of ROS, however the paper that they cite does not specifically state that?

Page 4 lines 172- 174 – please provide reference to support statement that ROS can shift T responses towards Th2?

Page 4, lines 182 -184 – please provide references for this statement – it feels that there are a lot of conclusions drawn from the data? If this is a speculation, then perhaps preface by saying “We speculate that ROS causes deficiency in Th1 responses and decreases IFNlevels, leaving individuals with asthma susceptible to virally-mediated induced exacerbations”

Figure 1 – recommended changing the title of the Figure 1 to “Summary of proposed mechanisms that contribute to oxidant/antioxidant balance in asthma pathophysiology.”

Page 6 – lines 217 – 219

The authors make a large assumption that decreased pH means increased 8-isopentate and increased oxidative stress.  This should be supported directly by literature or the authors should indicate that they are speculating that decreased pH means increased 8-isopentate and increased oxidative stress.

Page 10, lines 405-406, please provide a reference to support this statement 

Page 12, lines 489 -491 – this recommendation should be softened as people could take as medical advice.  It is suggested that the authors change the statement to indicate “Changes in diet, including increasing fruit and vegetable intake, along with regular PA may improve antioxidant status and respiratory health”

Figure 2 seems that it could be controversial and it is recommended that the authors eliminate the figure and/or change the title of the figure to “Proposed pharmacological and lifestyle interventions that might prevent oxidant/antioxidant imbalance .

Author Response

Comments and Suggestions for Authors from Reviewer#1

Introduction: 

Please provide literature/references to support the following statements:

 “In asthmatic patients, reactive oxygen species (ROS) production and consequent oxidative stress levels are elevated compared to healthy subjects. A vicious cycle of events supports the increased level of oxidative stress within the airways of asthmatics: external insults (allergens, pollution, smoke, etc) trigger oxidative stress events that, in turn, promote pro-inflammatory events, including the recruitment of activated inflammatory cells (eosinophils, neutrophils, and macrophages) that further increased oxidative stress since these cells are the primary source of ROS”

RESPONSE#1

We thank the reviewer for his/her comments. Following the reviewer’s suggestion, we have now cited references to support the statement regarding the increased levels of ROS in asthmatics.

Main Text

Page 3, the following sentence should be modified to specify which insults – for example, not all insults are predicted to activate toll-like receptors “The epithelial cells respond to external insults by activating innate responses (by Toll-like receptors (TLRs)), releasing defensins and mucus, coordinating local inflammation and the immune response through the release of cytokines and chemokines, activating antioxidant responses, and maintaining overall tissue homeostasis. “

RESPONSE#2

We thank the reviewer for his/her comment. Following the reviewer’s suggestion, we specified the type of insults and added references to support this.

Page 3, The following statement is not entirely correct.  The epithelial barrier is formed; however, the immune system is immature.  Please correct this statement accordingly: “At birth, the epithelial barrier is not fully formed, so in the first years of life, children are more vulnerable to external insults [23]”

RESPONSE#3

We have now amended as suggested by the reviewer.

Page 3, lines 112-113 - The authors suggest that cigarette smoke is one of the most significant sources of ROS, however the paper that they cite does not specifically state that?

RESPONSE#4

In the revised manuscript, we have now modified the text according to the reference.

Page 4 lines 172- 174 – please provide reference to support statement that ROS can shift T responses towards Th2?

RESPONSE#5

We have added the references in the revised manuscript.

Page 4, lines 182 -184 – please provide references for this statement – it feels that there are a lot of conclusions drawn from the data? If this is a speculation, then perhaps preface by saying “We speculate that ROS causes deficiency in Th1 responses and decreases IFNg levels, leaving individuals with asthma susceptible to virally-mediated induced exacerbations”

RESPONSE#6

Following the reviewer’s suggestion, we have now modified the sentence.

Figure 1 – recommended changing the title of the Figure 1 to “Summary of proposed mechanisms that contribute to oxidant/antioxidant balance in asthma pathophysiology.”

RESPONSE#7

We have now changed the title of Figure 1 as suggested by the reviewer.

Page 6 – lines 217 – 219

The authors make a large assumption that decreased pH means increased 8-isopentate and increased oxidative stress.  This should be supported directly by literature or the authors should indicate that they are speculating that decreased pH means increased 8-isopentate and increased oxidative stress.

RESPONSE#8

We have now amended as suggested integrating the references.

Page 10, lines 405-406, please provide a reference to support this statement 

RESPONSE#9

Following the reviewer’s suggestion, we have added references.

Page 12, lines 489 -491 – this recommendation should be softened as people could take as medical advice.  It is suggested that the authors change the statement to indicate “Changes in diet, including increasing fruit and vegetable intake, along with regular PA may improve antioxidant status and respiratory health”

RESPONSE#10

We have now amended the text as suggested by the reviewer.

Figure 2 seems that it could be controversial and it is recommended that the authors eliminate the figure and/or change the title of the figure to “Proposed pharmacological and lifestyle interventions that might prevent oxidant/antioxidant imbalance.

RESPONSE#11

Following the reviewer’s suggestion, we have now modified the title of figure 2.

Reviewer 2 Report

Dear Authors:

The manuscript "Oxidative stress, environmental pollution and lifestyle as determinants of asthma in children" by Di Vincenzo et al has summarized that the role of ROS in asthma of children. I have just a few suggestions.

Some references or background information is missing.  In introduction, please add more background information about ROS, which plays an important role in many diseases. It can emphasize the importance of your article. (Please cite:

1. An Epigenetic Role of Mitochondria in Cancer. Cells 2022, 11, 2518. https://doi.org/10.3390/cells11162518

2.    how far are we from truly understanding the pathogenesis of age-related dementia? Geroscience. 2022 Jun;44(3):1879-1883. doi: 10.1007/s11357-022-00591-7. 

3.  Mitochondrial mutations and mitoepigenetics: Focus on regulation of oxidative stress-induced responses in breast cancers. Semin Cancer Biol. 2022 Aug;83:556-569. doi: 10.1016/j.semcancer.2020.09.012.)

Best,

Author Response

Comments and Suggestions for Authors from Reviewer#2

Dear Authors:

The manuscript "Oxidative stress, environmental pollution and lifestyle as determinants of asthma in children" by Di Vincenzo et al has summarized that the role of ROS in asthma of children. I have just a few suggestions.

Some references or background information is missing.  In introduction, please add more background information about ROS, which plays an important role in many diseases. It can emphasize the importance of your article. (Please cite:

  1. An Epigenetic Role of Mitochondria in Cancer. Cells 2022, 11, 2518. https://doi.org/10.3390/cells11162518
  2. how far are we from truly understanding the pathogenesis of age-related dementia? Geroscience. 2022 Jun;44(3):1879-1883. doi: 10.1007/s11357-022-00591-7.
  3. Mitochondrial mutations and mitoepigenetics: Focus on regulation of oxidative stress-induced responses in breast cancers. Semin Cancer Biol. 2022 Aug;83:556-569. doi: 10.1016/j.semcancer.2020.09.012.)

Best,

RESPONSE

We thank the reviewer for his/her comments. We now included in the introduction the suggested references.

Reviewer 3 Report

Hello,

Thank you for submitting the manuscript titled “Oxidative stress, environmental pollution and lifestyle as determinants of asthma in children”. Authors have summarized the literature well and presented with figures.  Please see below minor comment.

Minor:

1.       There are some typos, spacing and bullets errors noticed in the manuscript. Please correct and proofread it.

Author Response

Comments and Suggestions for Authors from Reviewer#3

Hello,

Thank you for submitting the manuscript titled “Oxidative stress, environmental pollution and lifestyle as determinants of asthma in children”. Authors have summarized the literature well and presented with figures.  Please see below minor comment.

Minor:

  1. There are some typos, spacing and bullets errors noticed in the manuscript. Please correct and proofread it.

RESPONSE

We thank the reviewer for his/her comments. We have now further corrected and proofread the manuscript.

We hope that the revised manuscript is now suitable for publication in Biology.                                                               

Round 2

Reviewer 1 Report

The manuscript is strengthened and improved now